# Two-Stage Revision Arthroplasty for Resistant Gram-Positive Periprosthetic Joint Infections Using an Oral Linezolid-Based Antibiotic Regime

**DOI:** 10.3390/antibiotics12081235

**Published:** 2023-07-26

**Authors:** Lars Gründer, Sebastian Bockholt, Georg Gosheger, Martin Schulze, Jan Schwarze, Jan Pützler, Burkhard Moellenbeck, Christoph Theil

**Affiliations:** Department of Orthopedics and Tumor Orthopedics, Muenster University Hospital, Albert-Schweitzer-Campus 1, 48149 Muenster, Germany

**Keywords:** periprosthetic joint infection, prosthetic joint infection, PJI, revision arthroplasty, linezolid

## Abstract

Background: Increasing antibiotic resistance has been reported as an issue in the systemic treatment of periprosthetic joint infection (PJI). Linezolid offers the advantages of high oral bioavailability and little resistance; however, efficacy in the treatment of PJI varies considerably, and studies reporting consistent surgical treatment are scarce. Methods: This is a retrospective, single-center analysis of two-stage revisions performed between 2008 and 2017. We identified 111 patients who met the inclusion criteria. Oral linezolid was given for 28 days following 14 days of intravenous tailored antibiotics in resistant gram-positive PJI. A total of 64% of the patients had methicillin-resistant coagulase-negative staphylococci. The median follow-up was 43 (interquartile range (IQR) 30–57) months. Results: 22% (24/111) of the patients underwent surgery for subsequent infection. The 5-year infection-free survival probability was 77% (95% confidence interval (CI) 69–85). A total of 5% of the patients (6/111) had the same organism at the time of reinfection. The patients with infections caused by other organisms than *Coagulase-negative staphylococci* tended to have a worse reinfection-free survivorship at five years (70% vs. 81%, *p* = 0.09). Furthermore, the patients with obesity tended to have reduced reinfection-free survivorship at five years (69% vs. 84%, *p* = 0.08). Overall, 5% (6/111) of the patients had blood count abnormalities with no treatment discontinuations. Conclusion: Two-stage revision arthroplasty with systemic oral linezolid treatment for resistant gram-positive PJI results in an infection control of 77% at the mid-term.

## 1. Introduction

Periprosthetic joint infection (PJI) is a growing problem and a leading cause for the revision of total hip (THA) and total knee (TKA) replacements [1,2]. The prevalence of PJI in primary joint replacements is around 1–2% [3], while in patients who undergo revision arthroplasty it can be as high as 50% [4]. Given the continued high demand for joint replacement, the prevalence of revision arthroplasty for PJI is expected to rise. While surgical management depends on the timing, integrity of the implant, and soft tissue conditions, systemic antibiotics should be administered for a minimum of six weeks either intravenously or combined intravenously and orally [2,5,6]. Most PJIs are caused by gram-positive bacteria, most of them staphylococci [7,8]. Over the last years, the emergence of resistant gram-positive organisms, particularly methicillin-resistant coagulase-negative staphylococci or resistant enterococci [9,10], has greatly complicated systemic treatment, especially if oral continuation of therapy is desired [11,12]. Linezolid is a relatively new substance that can be used in the treatment of PJI either as an intravenous or oral formulation with nearly 100% oral bioavailability [13]. Current studies on resistance patterns have found that there is very little resistance against linezolid while resistance against other (orally) available antibiotics is increasing [14], although resistance to linezolid has also been reported in *enterococcus* infection [15]. Previous studies that have investigated the clinical use of linezolid in the treatment of PJI have found a successful eradication of infection in around 80% of cases depending on various factors [16,17,18,19,20]. However, previous studies were only able to include limited numbers of patients without usual follow-up periods with heterogeneous indications and surgical approaches limiting the clinical usefulness in the context of chronic PJI where a two-stage exchange is performed [16].

For instance, the largest studies on the use of linezolid-based antibiotic regimes [21,22] for PJI included around 50 patients, and the majority underwent surgery without implant retention in contrast to staged treatment.

Furthermore, a large proportion of studies included in a recent review used linezolid as a second-line treatment for recurrent infection or failure of previous systemic antibiotics, which is a substantial confounder for potentially lower treatment success [16].

This study aims at reporting clinical and microbiological results as well as treatment details regarding the use of linezolid for the treatment of PJI in a large cohort of patients treated with a standardized surgical approach and analyzes risk factors for treatment failure.

## 2. Results

### 2.1. Patient Details

Demographic details on the patients included are given in Table 1 and Table 2. Data on microbiology findings at first-stage surgery are given in Table 3.

### 2.2. Subsequent PJI

A total of 22% (24/111) of the patients underwent surgery for subsequent infection after second-stage reimplantation of the same joint after a median follow-up of 7 months (IQR 4–11). A total of 25% (6/24) of the patients had a causative organism from the same species, including polymicrobial infections, suggesting possible persistence (Table 4). A total of 58% (14/24) of the patients with subsequent PJI underwent repeat resection arthroplasty and staged revision, while the remaining 10 patients could be salvaged with implant retention, debridement, and antibiotics.

### 2.3. Microbiological Failure

A total of 5% of the patients (6/111) had the same species at the time of reinfection compared to the initial culture results, suggesting a microbiological failure of the two-stage approach with systemic linezolid. However, none of the organisms in reinfected patients had developed resistance to linezolid.

### 2.4. Infection-Free Survival

The 1-year infection-free survival probability amounted to 82% (95% CI 75–89), and the 5-year infection-free survival probability was 77% (95% CI 69–85) (Figure 1).

### 2.5. Factors Associated with Subsequent PJI

We found that the patients with infections caused by organisms other than ConS tended to have a worse reinfection-free survivorship at five years (70% vs. 81%, *p* = 0.09). Furthermore, the patients with obesity had a reduced reinfection-free survivorship at five years (69% vs. 84%, *p* = 0.08).

Additionally, the reinfected patients had a higher body mass index (BMI) (32 vs. 28, *p* = 0.002) and were of higher age (76 vs. 71, *p* = 0.08).

On the other hand, with the numbers available, there was no difference in infection-free survival depending on patient gender (*p* = 0.33), the type of intravenous systemic antibiotic treatment (*p* = 0.49), the type of local antibiotic added to the spacer (*p* = 0.5), diabetes (*p* = 0.24), McPherson systemic host grade (*p* = 0.58), McPherson extremity grade (*p* = 0.29), or the presence of fistulating infection (*p* = 0.64). In addition, the patient’s CCI (*p* = 0.68) and the number of previous surgeries (*p* = 0.13) were not associated with the risk of subsequent PJI.

### 2.6. Adverse Events and Treatment Discontinuation

There were no treatment discontinuations reported. Overall, 5% (6/111) of the patients had blood count abnormalities, and 3 patients had thrombopenia and leukopenia.

## 3. Discussion

Increasing antibiotic resistance particularly among CoNS PJI is an increasing challenge for clinicians [10]. Furthermore, in order to reduce hospital stays and lower healthcare costs, outpatient oral treatment appears desirable. Linezolid offers the advantage of little resistance reported and almost 100% bioavailability [2,20]. While previous studies have evaluated the use of linezolid-based antibiotic regimes in implant retention approaches with reasonable patient numbers included [19], linezolid use in chronic infections with a two-stage revision arthroplasty approach including follow-up data are scarce and limited by small numbers [25,26]. This study finds an infection control rate of 78% and infection-free survivorship of 77% after five years. Furthermore, obese patients with infections caused by organisms other than CoNS were at increased risk for recurrent infection.

The likelihood of successful infection control using a two-stage approach, including systemic linezolid treatment, has been investigated previously. One study [25] included twelve patients with similar baseline characteristics in terms of comorbidities and causative organisms and found a 100% cure rate. Considering the various factors that can potentially influence the outcome of PJI treatment and the small number of patients included in their study, it is unclear what explains the difference in survival compared to the present study. Considering that the reinfection risk for uncomplicated first-time PJI of a TKA or THA [27,28] amounted to 10–15%, it appears plausible that complicated PJI with resistant gram-positive organisms might have a higher risk of reinfection, as reported in the present investigation.

Furthermore, Eriksson et al. [18] investigated 28 patients who had PJI caused by CoNS, and were 15 (54%) who were managed using a two-stage approach. They found an infection control rate of 87% with a similar long-term follow-up as the present study. However, unlike the present study, they used antibiotic suppression therapy frequently, which might make it difficult to directly compare the studies. Surgeons should, therefore, be aware that even with standardized antibiotic treatment with oral linezolid for resistant gram-positive infections, the outcome of treating PJI with a two-stage exchange may vary tremendously with a number of factors and may have an impact on survival.

Regarding microbiology findings, we found that the most common organism cultured was resistant coagulase-negative staphylococci, and the patients with a different organism than CoNS had a higher risk of relapse in comparison. The reasons for this are unclear given the retrospective study design. However, PJI caused by resistant CoNS is a major concern, although resistance patterns appear to vary tremendously, even in similar geographical regions [29]. One study on 55 episodes of TKA PJI caused by CoNS found that only around half of the patients, despite the use of long-term systemic antibiotic treatment, remained infection-free after a follow-up period of 29 months [30]. Compared to the present study, 40% of their patients had undergone treatment for PJI before and in their study, this was associated with a higher risk of reinfection. Interestingly, resistant CoNS were not associated with worse infection control in their study. Considering the diverging results, future studies should comprehensively investigate standardized surgical management as well as uniform systemic therapies, including the route of administration and duration.

In the present study, patients with obesity had a higher risk of reinfection compared to patients with normal body weight. This is in line with findings from previous studies on two-stage revision. One study [31] included 97 episodes of hip PJI found a significantly reduced infection-free survivorship of 60% vs. 80% at five years for patients with obesity. Considering these findings, obese patients with resistant gram-positive PJI should be considered high-risk patients and counseled accordingly.

While this is not a study on drug safety, we noted no discontinuations of oral linezolid treatment, and there were only 5% blood count abnormalities in the present cohort. In this study, linezolid was used for treatment continuation after intravenous treatment, and treatment duration was limited to 28 days, according to the pharmaceutical companies’ recommendations. Nonetheless, considering that longer treatment durations for PJI are desirable and may lead to favorable outcomes, including lifelong suppressive therapy in high-risk cases [20,32,33,34,35,36], surgeons might consider linezolid for longer applications [37]. Nonetheless, considering the reported high rates of adverse drug reactions in up to 40% of cases [16,33], therapeutic drug monitoring and close general surveillance appear warranted. Furthermore, an optimal antibiotic drug combination is still being discussed, as there is a potential for interactions and insufficient dosing of linezolid, particularly for rifampicin as an antibiofilm agent [38,39].

While this is a large study with a relatively homogeneous patient cohort, its findings must be interpreted considering several limitations. First, it is retrospectively designed, and it is possible that some patients might have undergone treatment for PJI elsewhere; therefore, the reported infection risk should be viewed as a low-end result. Nevertheless, the median follow-up is almost four years, which exceeds many previous studies on PJI treatment with linezolid. Secondly, when analyzing risk factors for infection recurrence or treatment failure, this study is limited by the number of patients with individual risk factors. Furthermore, some risk factors, such as individual culture results or patient factors, might be present, but the study could still be underpowered to detect those. Thirdly, while we feel confident to report on some adverse events that could be attributed to linezolid treatment, this study is not a pharmaceutical or prospective trial., Patients were not prospectively supervised and the majority had blood counts performed on an individual basis by the general practitioner. Considering the higher risk of adverse events reported in previous studies, claims regarding drug safety should not be overstated based on our results.

Fourthly, the outcome of two-stage chronic PJI depends on various factors, and systemic antibiotics are only considered as part of a multimodal treatment approach considering various factors, including intravenous systemic and local antibiotics. While we absolutely acknowledge these factors and their potential impact on our results, this study was unable to detect an effect on the survival and eradication of the infection based on the systemic antibiotic used or the presence of vancomycin in the spacer. Furthermore, multivariate analyses with larger datasets providing hazard ratios are needed to elucidate the potential interdependencies.

## 4. Materials and Methods

### 4.1. Study Design

After obtaining approval from our local ethics committee (reference number F-s-2019-041-f-S), we conducted a retrospective database search of our institution’s electronic database and reviewed all patients that underwent revision surgery of an infected hip or knee joint replacement between 2008 and 2017. Overall, 625 patients (275 THA, 350 TKA) completed a two-stage revision for chronic PJI, and 514 did not undergo treatment with linezolid.

We included patients who underwent non-oncologic joint replacement as well as patients who underwent implant revision following oncological resection and implantation of tumor prostheses. We did not review patients with spinal infections, infected fractures, or osteosynthesis and osteomyelitis without foreign material present. Applying those criteria, we included 111 patients with a two-stage hip or knee revision for PJI with resistant gram-positive bacteria undergoing systemic linezolid treatment.

In order to diagnose infection, we applied the criteria of the musculoskeletal infection society (MSIS) of 2011 [40], and prior to that the Centers for Disease Control (CDC) criteria [41]. Synovial and tissue cultures were at least cultured for 7–14 days on Columbia blood agar, Schaedler agar, and chocolate agar. Resistance testing was performed based on the EUCAST (European Committee on Antibiotic Susceptibility Testing) recommendations of the respective year. For chronic infections, a two-stage approach was performed with first-stage removal of all foreign material and debridement, which included the resection of all infected, necrotic, and fibrous tissue, bone sequestra, and inflammatory synovium. In particular, the debridement of the acetabular cavity and intramedullary cavities with sequential reaming using a flexible reamer was performed. A chronically infected bone that showed no signs of bleeding after debriding was resected. All wounds underwent irrigation using pulsatile lavage on alternating high- and low-pressure settings. Ultimately, the insertion of an antibiotic-loaded polymethylmethacrylate (PMMA) handmade spacer (Copal G + C, Heraeus Medical GmbH, Wehrheim, Germany) was performed. For revision TKA, we utilized a static spacer design using two 6 mm titanium rods that were cut to length and coated with cement. The joint space was subsequently filled with the remaining antibiotic-loaded cement. For revision THA, an articulating spacer was formed matching the acetabular and femoral diameter to achieve adequate stability and reconstruct the femoral offset. Topical antibiotics were added based on resistance testing with 2 g of gentamicin and 1 g of clindamycin added per 40 g of cement for all sensitive bacteria. For culture-negative infections or resistant gram-positive infections, a vancomycin spacer (2 g per 40 g of cement) was added (80% of patients (89/111). We did not use other antibiotic carriers during the study period.

Systemic antibiotics were given for a minimum of six weeks and were generally administered every two weeks as an intravenous treatment, and the oral continuation treatment was performed in an outpatient setting for 4 weeks. In two-stage exchanges, intravenous therapy was given after reimplantation surgery for two further weeks until the final culture results from the reimplantation surgery remained negative. Otherwise, another four weeks of oral antibiotics were given.

Failure was defined as a patient requiring further surgery due to infection. Furthermore, microbiological failure was defined as revision surgery for infection caused by the same species. On the other hand, infection control was defined as proposed by the consensus criteria by Diaz-Ledema et al. [42], which required healed wounds, no revision surgery for infection, and the absence of mortality due to PJI.

Patient demographic details (Table 1 and Table 2) and microbiological findings (Table 3) were extracted from our institution’s electronic patient records. Based on the recorded previous medical history, we calculated the Charlson comorbidity score (CCI) for each patient and gathered information regarding previous surgeries. Furthermore, the host and extremity grade according to McPherson was calculated for all patients [23,24]. This system uses a combined host grade and extremity grade in order to give an estimate regarding the risk of complications in revision arthroplasty for infection. The follow-up was calculated from the revision surgery until the last contact with our institution or the patient’s death. The median follow-up was 43 (IQR 30–57) months.

In the present study, linezolid was used as an oral, outpatient continuation of previous intravenous therapy due to its excellent oral bioavailability and was administered for 28 days (maximum length of therapy according to the manufacturer). In some cases of infection caused by vancomycin-resistant enterococci, linezolid was already used during the hospital stay (2%, 3/111). Intravenous treatment was performed using vancomycin in 72% of the patients (80/111), flucloaxacillin or cephalosporines in 18% (20/111), and daptomycin in 7% (8/111).

Linezolid was used as the first choice to treat resistant gram-positive infections if other oral alternatives (quinolones or cotrimoxazole) were tested to be resistant or if comorbidities or coexisting medications taken by individual patient did not allow for other oral antibiotics (five patients for cotrimoxazole and seven patients for leveofloxacin). A total of 600 mg of linezolid was given twice a day.

A total of 14% of the patients (15/111) had combined oral therapy due to polymicrobial growth, and 10 patients received oral amoxicillin/clavulanic acid and 5 patients received ciprofloxacin.

A combination with rifampicin was not generally used during spacer treatment because of relevant interaction and a reduction in linezolid blood levels caused by rifampicin [43,44,45]. We did not use linezolid in patients who relied on continuous therapy with serotonin reuptake inhibitors due to the risk of serotonin syndrome. We did not use suppression therapy during the study period.

The patients had a blood count performed after systemic antibiotic therapy during outpatient assessment prior to second-stage surgery and were questioned regarding treatment discontinuation or other complications.

### 4.2. Statistical Analysis

A descriptive data analysis was conducted, and the distribution of the data was analyzed using the Shapiro–Wilk test or the Kolmogorov–Smirnov test. Parametric and non-parametric analyses were performed using Student’s *t*-test or the Mann–Whitney u-test, respectively. Categorical variables were analyzed with the chi-squared test using cross tables. The reinfection-free implant survivorship was analyzed with the Kaplan–Meier method, and differences in survivorship were compared with the log-rank test. P was set at 0.05; all *p*-values were two-sided. The respective 95% confidence intervals (CIs) were calculated.

## 5. Conclusions

A treatment approach using two-stage revision arthroplasty in combination with a systemic antibiotic regime consisting of intravenous treatment followed by oral linezolid treatment for resistant gram-positive PJI resulted in an infection control of 77% at mid-term follow-up with a low rate of discontinuation and adverse events. Future studies should investigate optimal combination therapy as well as the prolonged use of linezolid.

## Figures and Tables

**Figure 1 antibiotics-12-01235-f001:**
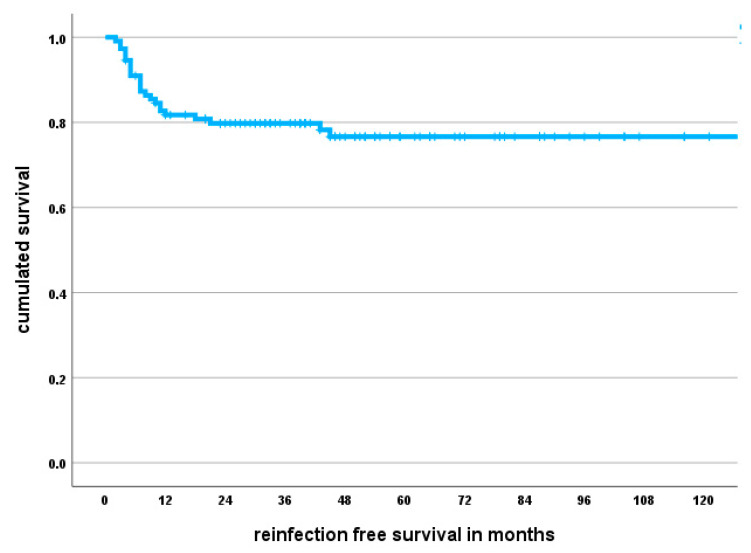
Kaplan–Meier survival curve showing infection-free survival probability. The median follow-up amounted to 43 months; however, individual patients were followed for up to ten years.

**Table 1 antibiotics-12-01235-t001:** Patient demographics.

Variable	Median (25–75% IQR)	Reinfection Group (24/111)	Infection Control Group (87/111)	*p*-Value
Age	72 (63–78)	76	71	0.08
CCI	3 (1–4)	4	3	0.68
BMI	29 (26–34)	32	28	0.002
Number of previous revision arthroplasties	3 (1–5)	2	3	0.38
Days until second-stage reimplantation	67 (52–82)	75	64	0.13

**Table 2 antibiotics-12-01235-t002:** Patient demographics, including host and extremity grade [23,24].

Variable	% (n/111)	Reinfection Group % (n/24)	Infection Control Group % (n/87)	*p*-Value
Female	56 (62)	46 (11)	59 (51)	0.33
Diabetic	24 (27)	8 (33)	22 (19)	0.24
Implant location				0.82
THA	48 (53)	50 (12)	47 (41)	
TKA	52 (58)	50 (12)	53 (46)	
Currently smoking	3 (3)	n/a	3 (3)	n/a
Obesity (BMI > 30)	46 (51)	63 (15)	41 (36)	0.05
Previous revision arthroplasty	87 (96)	75 (18)	90 (78)	0.13
Previous revision for PJI	53 (59)	50 (12)	54 (47)	0.82
Fistulating infection	22 (24)	17 (4)	23 (20)	0.64
Systemic host grade				0.58
A	50 (56)	46 (11)	52 (45)	
B	47 (52)	50 (12)	46 (40)	
C	3 (3)	4 (1)	2 (2)	
Extremity grade				0.29
1	8 (9)	n/a	10 (9)	
2	72 (80)	83 (20)	69 (60)	
3	20 (22)	17 (4)	21 (18)	

**Table 3 antibiotics-12-01235-t003:** Microbiology at first-stage surgery. * *ConS: coagulase-negative staphylococci*; *MR-ConS: methicillin-resistant coagulase-negative Staphylococci*; *MRSA: methicillin-resistant staphylococcus aureus*; *MSSA: methicillin-sensitive staphylococcus aureus.* ** *Corynebacterium*, *finegoldia*, and *peptostreptococcus* >100% due to polymicrobial infections with ConS and others listed above.

Microbiology at First-Stage Surgery *	% (n/111)
*MR-ConS*	64 (71)
*MRSA*	11 (10)
*MSSA*	8 (9)
*Enterococcus*	7 (6)
*Polymicrobial*	33 (37)
Others **	13 (15)

**Table 4 antibiotics-12-01235-t004:** Microbiology at reinfection, * candida, streptococci (n = 2), staphylococcus aureus, enterococcus.

Microbiology at Reinfection	% (n/24)
Polymicrobial	42 (10)
*MR-coagulase negative staphylococci (CoNS)*	21 (5)
Gram-negative microorganisms	17 (4)
Others *	21 (5)

## Data Availability

Anonymized datasets are available upon reasonable request.

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
