# Peer review of "Two-Stage Revision Arthroplasty for Resistant Gram-Positive Periprosthetic Joint Infections Using an Oral Linezolid-Based Antibiotic Regime"

_antibiotics, 2023, doi:10.3390/antibiotics12081235_

Round 1

Reviewer 1 Report (New Reviewer)

This single-center, retrospective study describes the use of a linezolid (LZD) as a follow-up treatment  in 111 patients with PJI, over a  9-year period. It brings new data to the present literature, which relies mainly on similarly-conducted retrospective studies.

The manuscript is well-written and easy-to-read. Data are presented clearly. However I have some remarks and questions:

- Line 29 and 126-130. It is commonly admitted that a difference may be considered as significative when p<0.05. All differences in the manuscript with p>0.05 should be considered “tendencies”.

- Table 4. The emergence of resistance to LZD is not mentioned, are there data in this institution which commonly uses LZD ? Was it observed in reinfections, especially when a similar bacterial species was involved ?

- Table 1. These patients seem to have undergone  several revision arthroplasties before the present one (betwenn 1 and 5), were they mainly for infections? Mechanical failures? In other terms, did the patients suffered several bacterial infection episodes (and received multiple lines of ATB therapy) before the one investigated here?  What was the % patients with previous surgeries?

- Table 3. The authors state (line 306) that LZD was mainly used in case of cotrimoxazole/quinolones resistance. Were all these strains resistant to both these ATB classes? Moreover, which patients also received fluoroquinolones or co-amoxiclav, as mentioned at line 309 (as both these ATBs may sometimes have activity against MSSA or enterococcus)?

- It would have been interesting to know if patients over the same time period were treated with another follow-up treatment in this institution (e.g. with cotrimoxazole of quinolones), and the % of success when compared to LZD follow-up treatment. Do the authors have this data?

- Line 103. What were the criteria used to define “same organism”? Same species? Same ATB resistance phenotype?

- Table 4. “polymicrobial” seems to cover additional bacteria (other than CoNS, Gram-negative, or “others”); what were they?

-Figure 1. When reading the text I did not understand that patients were followed-up until over 10 years (120 months)?

- Line 145. Do the authors have data on the median time of occurence of thrombopenia and leukopenia, and the median platelet and PMN cells lowest  counts?

- Line 249. A little more information on patient selection would be wellcome (maybe in a flowchart?): Number of patients screened? Excluded? REasons for exclusion, Number with other ATB treatments?

- I did not see a mention of the median delay between both stages of the two-stage revision arthroplasty?

- Were all ATB treatments given for strictly six weeks (14 days IV ATBs + 28 days LZD)? If not, more information should be given on LZD treatment duration (mean, median, ranges).

- Line 277. “In two-stage exchanges”: Why this precision, since I believed that all 111 patients underwent two-stage exchanges (it is the title of the study)?

Minor comments.

- Table 2. Current smoking: 3/87 does not make 100%

- Some minor typo and grammar mistakes should be checked over.

Some minor typo and grammar mistakes should be checked over.

Author Response

Dear reviewer, thank you for your thoughtful and comprehensive revision of our manuscript.

We agree with you that several aspects would benefit from clarification and more details. We have rewritten several parts of the manuscript to accomodate for the changes made:

Regarding the specific comments:

Line 29 and 126-130. It is commonly admitted that a difference may be considered as significative when p<0.05. All differences in the manuscript with p>0.05 should be considered “tendencies”.

Thank you, we have changed this to tended or tendency at all instances.

Table 4. The emergence of resistance to LZD is not mentioned, are there data in this institution which commonly uses LZD ? Was it observed in reinfections, especially when a similar bacterial species was involved ?

Thank you very much, luckily no patient developed LZD resistance. We have added this to the results. This is an important addition. Furthermore, we have added to the introduction that resistance has unfortunately been reported. We have added to line 57 and added reference 15.

Table 1. These patients seem to have undergone  several revision arthroplasties before the present one (betwenn 1 and 5), were they mainly for infections? Mechanical failures? In other terms, did the patients suffered several bacterial infection episodes (and received multiple lines of ATB therapy) before the one investigated here?  What was the % patients with previous surgeries?

Thank you very much, this information is in table 2 as we did not wish to mix metric and catergorical variables in one table. 87% of patients have undergone previous revision and 57% had previous surgery for PJI.

Table 3. The authors state (line 306) that LZD was mainly used in case of cotrimoxazole/quinolones resistance. Were all these strains resistant to both these ATB classes? Moreover, which patients also received fluoroquinolones or co-amoxiclav, as mentioned at line 309 (as both these ATBs may sometimes have activity against MSSA or enterococcus)?

Thank you very much. We have added to lines 312ff.  In 12 patients total cotrimoxazole or levofloxacin were a potential option, however due to intolerance, co-medication or medical history LZD treatment was recommended. 

It would have been interesting to know if patients over the same time period were treated with another follow-up treatment in this institution (e.g. with cotrimoxazole of quinolones), and the % of success when compared to LZD follow-up treatment. Do the authors have this data?

Thank you very much, while we absolutely agree with you that a comparative study design would be helpful, but we are currently collecting this data as it is a somewhat heterogeneous and complex study population.

Line 103. What were the criteria used to define “same organism”? Same species? Same ATB resistance phenotype?

Thank you very much, we have changed this to same species which is what we meant. In these cases however, they had the same resistance pattern as well.

Table 4. “polymicrobial” seems to cover additional bacteria (other than CoNS, Gram-negative, or “others”); what were they?

Thank you very much ,among "others", we summed up Corynebacteria, Finegoldia and peptostreptococcus. These percentages do not add up, as patients with polymicrobial infectiosn might have had CoNS and were counted for bothe lines. Please apologize the confusion. We have added this to the Table 3 caption.

-Figure 1. When reading the text I did not understand that patients were followed-up until over 10 years (120 months)?

Thank you very much, the median follow-up was 43 months and we chose to use non-parametric measures as the data was not distributed normally. The survival curve shows the end of follow-up for an individual patient that was almost 10 years.

Line 145. Do the authors have data on the median time of occurence of thrombopenia and leukopenia, and the median platelet and PMN cells lowest  counts?

Thank you very much, due to the retrospective design, we do not have the precise time point during the  28 day treatment. From personal experience, many patients in whom we do an inpatient monitoring have some drop during the first couple of days and another few after completing the time.

Line 249. A little more information on patient selection would be wellcome (maybe in a flowchart?): Number of patients screened? Excluded? REasons for exclusion, Number with other ATB treatments?

Thank you, we have added the number of completed two-stage revision during that time period who did not undergo linezolid treatment. Unfortunately, there has not been a detailed work-up on these cases (yet) which makes it difficult to establish a control group. Maybe for future studies.

I did not see a mention of the median delay between both stages of the two-stage revision arthroplasty?

Thank you, we have added this to table 1.The median interval was 67 days.

Were all ATB treatments given for strictly six weeks (14 days IV ATBs + 28 days LZD)? If not, more information should be given on LZD treatment duration (mean, median, ranges).

As part of this algorithm, antibiotic duration was timed with 14 days i.v. and 28 days of linezolid. Particularly for linezolid, during the study period, we did not use it for prolonged (in Germany off-label) therapy longer than 28 days.

- Line 277. “In two-stage exchanges”: Why this precision, since I believed that all 111 patients underwent two-stage exchanges (it is the title of the study)?

Thank you, this is unncessary and has been removed.

Minor comments.

- Table 2. Current smoking: 3/87 does not make 100%

  • Some minor typo and grammar mistakes should be checked over.

Thank you, this has been corrected and we had a native speaker proof read the manuscript.

Thank you again for your thoughtful and very helpful review.

Kind regards

Reviewer 2 Report (New Reviewer)

1. How was the sample size estimated in this study?

2. Please estimate the hazard ratios with 95% confidence intervals for the reinfection rates.

Professional edits for language is necessary.

Author Response

Dear reviewer,

thank you for your comments on how to improve the manuscript that we are happy to adress:

  1. While we absolutey agree that sample size calculation is important in planning good quality studies, we would like to point out that this is a retrospective study. We included all patients who underwent two-stage exchange for PJI using systemic oral linezolid treatment over a 9 year period. As the study was formulated post hoc, there is no way to provide a sample size calculation. We have emphasized in the limitations that some factors in comparative analysis might be affected by sparse data bias and this study might be underpowered. H
  2. Thank you very much for the comment. We have provided 95% Confidence intervals for the survival estimates using the Kaplan-Meier analysis. Hazard ratios would be a metric to results from multivariate analysis/comparative analysis. Considering that many factors are associated with reinfection and even with univariate comparisons, we were unable to find significant differences in the likelihood of reinfection, our statisticians advised against multivariate/Cox regression analyis and therefore there are no hazard ratios. We have pointed this out in the limitations.
  3. The manuscript has been proof-read by a native speaker for language.

We hope that our comments and the additions to the limitations qualify our manuscript for publication

Round 2

Reviewer 1 Report (New Reviewer)

Corrections were made and more details were given by the authors. I see two remaining minor points to be addressed:

Line 103. What were the criteria used to define “same organism”? Same species? Same ATB resistance phenotype?

Thank you very much, we have changed this to same species which is what we meant. In these cases however, they had the same resistance pattern as well.

=> The change does not appear in the version of the manuscript (line 105).

 -Figure 1. When reading the text I did not understand that patients were followed-up until over 10 years (120 months)?

Thank you very much, the median follow-up was 43 months and we chose to use non-parametric measures as the data was not distributed normally. The survival curve shows the end of follow-up for an individual patient that was almost 10 years.

=> This detail should be written somewhere (maybe as a footnote to the figure?) in order not to puzzle the reader.

Author Response

Dear reviewer,

thank you very much for the important comments. We have made both corrections.

Kind regards

Reviewer 2 Report (New Reviewer)

Thank you for the revision.

Minor edits are required.

Author Response

Thank you very much.

The manuscript has been revised by a native english speaker and we did not detect any editable flaws at this point. However, of course we are happy to make any changes to the English style used during editing or if you provide any examples on how to reword.

Kind regards

This manuscript is a resubmission of an earlier submission. The following is a list of the peer review reports and author responses from that submission.

Round 1

Reviewer 1 Report

I read the manuscript “Two-stage revision arthroplasty for resistant gram-positive periprosthetic joint infections using an oral linezolid based antibiotic regime” by Gründer et al, candidate for publication in Antibiotics. Although the effort of the authors to review all cases and analyze the findings is to be acknowledged and celebrated, my impression is that the manuscript, in its present form, is not suitable for publication. The article includes a single-center experience on the use of linezolid for PJI managed with a two-stage exchange surgery in 111 patients. The main weakness of the article, from my perspective, is the lack of originality and novelty of the data included in the analysis, as there is wide experience in the use of linezolid for such infections. Although it is true that there are not many case-series of implant removal and linezolid treatment, the data included in the present manuscript includes very little information regarding the use of linezolid itself. Possibly, more information on treatment data or even a comparison with treatment alternatives would improve the manuscripts.

Additionally, while the introduction is well written, the methods section lacks important information. There are no definitions of failure; how was it defined? I understand from the results section that any reinfection was considered as failure, but when evaluating the outcome accountable for linezolid, I believe that it would be more appropriate to focus on microbiological failure. This occurred to 6 patients (5%), which is similar to general cohorts of patients with PJI and managed with implant removal, and far from the 22% suggested in the study as overall failure. We also don’t know if failure could occur at any time since first-stage exchange or was considered after reimplantation. Were mortality or suppressive treatment considered as failures? Later on results, a mention to infection-free survival is made. How is this defined? How is it different from failure?

I would also suggest to review the statistics used in the study. There is a mention of a Kaplan Meier analysis in methods, not shown in results. In Lines 91-93, I think that confidence intervals do not make much sense here, when comparisons are made. In the results section, the data is poorly presented and a bit chaotic. There is no formal presentation of the cohort, which is found in Table 2 and referred in Methods (I suggest you move to Results section). Table 2 and 3 may be merged as well. IQR are not percentages (see Table 2). Table 4 refers to microbiology of the first surgery and appears after the microbiology of the outcome. There is no table on the comparison between patients who failed and who did not. The data included on adverse events or other characteristics of linezolid treatment is poor; when did adverse events happen? Interactions with antidepressant medications?

I believe that the manuscript would be strengthened if some of this points are addressed. Other points:

- Please review capital letters in Tables. Gram-negatives should be “Gram-negative microorganisms”, for example. Microorganism names should be in italics.

- Lines 194-196. Sentence ending in “who” does not make much sense. Please review.

- Lines 206-208. Consider removing this info on implant retention as your manuscript only includes two-stage exchange.

- Lines 211-214. How did you know the resistance testing prior to surgery? Had microbiological samples before surgery? Clindamycin resistance is high in CoNS, have you considered changing to vancomycin in all patients?

- Line 247. Quinolones.

Some sentences need improvement.

Author Response

Dear reviewer,

thank you very much for your thorough and extremely helpful review. We greatly appreciate the effort that you put into making suggestions on how to improve the manuscript and to provide a clearer, more differentiated perspective to the readers on linezolid treatment in two-stage revision for PJI.

We have changed several paragraphs and incorporated your suggestions.

Regarding specific comments made:

is the lack of originality and novelty of the data included in the analysis, as there is wide experience in the use of linezolid for such infections

Thank you for your comment, while we agree with the general notion that linezolid use in PJI has been studied before, there is a lack of homogeneus studies on two-stage revisions for chronic PJI. Furthermore, this is by far the largest study on clinical success. We therefore consider the mansucript novel and original in that sense. The drawbacks of a retrospective design have been adressed in the limitations.

Possibly, more information on treatment data or even a comparison with treatment alternatives would improve the manuscripts.

Thank you very much, we absolutely agree with you that we need comparative, prospective trials in order to find an optimal systemic treatment in terms of substance choice, duration etc. Particularly for resistant gram positive organisms, there is no gold standard. Considering that this is a retrospective study, we lack such a control group and have amended the limitations sections on that matter.

There are no definitions of failure; how was it defined?

Thank you very much, we have added this to the methods, we used the consensus definition by Diaz-Ledema et al. that is a clinical definition. 

 it would be more appropriate to focus on microbiological failure

Thank you, this is an important aspect if we consider antibiotic treatment succes. We have added a paragraph to the results on microbiological failure that we defined in the methods section. We considered reinfection with the same organism as a microbiological failure.

 Were mortality or suppressive treatment considered as failures

Thank you very much for pointing out this important aspect. We did not use suppression treatment during the study period, this was added to the methods section. Mortality due to PJI was considered a failure according to the definition by Diaz-Ledema et al. Mortality in general if not directly related to PJI (e.g. septic shock etc.) was not considered a failure. There is a lot of ongoing work currently regarding the definition of success after PJI treatment and novel studies have found stark differences in success rates according to definitions used (PMID:33543876, Borsinger et al. CORR 2021)

Later on results, a mention to infection-free survival is made. How is this defined? How is it different from failure?

Thank you very much. We used absolute measures (rate of reinfection) in absolute numbers and percentages.. Furthermore, we used Kaplan Meier survival estimates in order to calculate the likelihood of failure at certain time points considering the differences in follow-up.

In Lines 91-93, I think that confidence intervals do not make much sense here, when comparisons are made

Our statistical advisors told us that confidence intervals should be reported and the editors at MDPI had asked for CIs in past publications.

here is no table on the comparison between patients who failed and who did not.

Thank you very much ,we have added to tables 1 and 2 to display the characeteristics of patients who failed (failure defined as revision for infection)

The data included on adverse events or other characteristics of linezolid treatment is poor when did adverse events happen? Interactions with antidepressant medications?

Thank you very much, the adverse events were detected during the 28 days of outpatient treatment and upon outpatient check up following this time. We did not use linezolid in patients who were on serotonine reuptake inhibitors and were unable to discontinue those for a short period. However, it is important to keep the risk of serotonine syndrom in mind. 

- Please review capital letters in Tables. Gram-negatives should be “Gram-negative microorganisms”, for example. Microorganism names should be in italics.

 Thank you, this has been changed.

- Lines 194-196. Sentence ending in “who” does not make much sense. Please review.

  Thank you, this has been changed.

- Lines 206-208. Consider removing this info on implant retention as your manuscript only includes two-stage exchange.

 Thank you, this has been changed.

- Lines 211-214. How did you know the resistance testing prior to surgery? Had microbiological samples before surgery? Clindamycin resistance is high in CoNS, have you considered changing to vancomycin in all patients?

 Thank you, patients underwent aspiration for microbiology culture prior to surgery or had culture results from revision surgeries at outside hospitals. We absolutely agree with you that when in doubt if organisms might have become more resistant compared to previous results or in culture-negative infections we nowadays use Vancomycin in the spacer empirically.

  • Line 247. Quinolones.

This has been changed thanks.

Thank you again for your important comments and your work on the review.

Kind regards

Reviewer 2 Report

Absract:

- Even in abstracts, abbreviations must be written out the first time they are mentioned, e.g. PJI and CoNS

Introduction:

- There is an issue with the brackets and spacing before brackets throughout the manuscript, e.g. see lines 53,55,57,66.

Results:

- line 91,93: again a similar issue with brackets

- line 92-100: here are several abbreviations that are not explained. It might be clear to almost everybody what BMI means, you still have to wright it out the first time they are mentioned. Furthermore, what is McPherson? This should be explained in the methods section. Lastly, you use two different words here “survival” and “survivorship”. Why? Is this on purpose? If yes – explain, if not – change.

- line 105: patients themselves did not report thrombopenia.

Discussion:

- I would recommend putting limitations at the end of the section

Methods:

- line 196: “… as well as patients who.” What happened here? Change accordingly.

- line 202: “(CDC)” has to come before the word “criteria”.

The major issue I see is that you do not report what the exact intravenous antibiotic treatment was.

Furthermore, was linezolid given combined in some cases or not?

Furthermore, the order of the manuscript sections is very strange. If this is not required by the journal, then you must put the method section after the introduction section.

Author Response

Dear reviewer,

thank you very much for your important comments that will help to improve the quality of the manuscript in order to become a valuable contribution to the scarce literature on linezolid treatment for PJI. We hope that changes in the manuscript will help to bring the message out more clearly.

Regarding the specific comments made:

Even in abstracts, abbreviations must be written out the first time they are mentioned, e.g. PJI and CoNS

Thank you, this has been changed throughout.

Introduction:

- There is an issue with the brackets and spacing before brackets throughout the manuscript, e.g. see lines 53,55,57,66.

Thank you, this has been changed. This appears to be an issue with the template and the citation/literature software. I hope that this will not occur again.

Results:

  • line 91,93: again a similar issue with brackets

Thank you, this has been changed accordingly.

  • line 92-100: here are several abbreviations that are not explained. It might be clear to almost everybody what BMI means, you still have to wright it out the first time they are mentioned. Furthermore, what is McPherson? This should be explained in the methods section. Lastly, you use two different words here “survival” and “survivorship”. Why? Is this on purpose? If yes – explain, if not – change.

Thank you, this has been changed accordingly. We have also expanded the methods section to accomade an explanation for the McPherson grading system that allows to define a host and local extremity grade in order to give an estimate on how compromised the extremity and the patient's overall health are. We changed everything to survivorship as this appears to be the more accurate term from a statistical point of view.

- line 105: patients themselves did not report thrombopenia.

This incorrect wording has been changed. Thank you.

Discussion:

- I would recommend putting limitations at the end of the section

Thank you, this has been changed.

Methods:

  • line 196: “… as well as patients who.” What happened here? Change accordingly

This has been changed, thank you.

line 202: “(CDC)” has to come before the word “criteria”.

Thank you this has been changed.

The major issue I see is that you do not report what the exact intravenous antibiotic treatment was.

Thank you this has been changed and added to the methods.

Furthermore, was linezolid given combined in some cases or not?

Thank you, this is an important aspect. While linezolid combinations are not easy from a pharmaceutical standpoint, in mixed gram-negative infections this can't be avoided. In these cases, an oral aminopenicillin or ciprofloxacin were added depending on comedication. We have added to the methods sections as well regarding this aspect. 

Thank you again for your important comments and the thorough work on the paper.

Kind regards

Furthermore, the order of the manuscript sections is very strange. If this is not required by the journal, then you must put the method section after the introduction section.

Reviewer 3 Report

Excellent study on Two Stage revision arthroplasty for resistant gram positive periprosthetic joint infections using an oral linezolid based antibiotic based regime.

Good introduction  and background to morbidity and mortality associated with prosthetic joint infection.

Methods described in manuscript is appropriate.

Good discussion and results showing benefit of linezolid based regime in preventing recurrent PJI ion revision surgery

Conclusion is supported by results

As above. Good quality paper acceptable for publication

Author Response

Dear reviewer,

thank you for your very appreciative review.

While I do not see any suggestions for improvement based on your comments, we have added some more information to the methods section and rewritten parts of the results and introduction based echoing the comments of the other reviewers.

Thank you again for your work and kind regards

Round 2

Reviewer 1 Report

I read again the manuscript “Two-stage revision arthroplasty for resistant gram-positive periprosthetic joint infections using an oral linezolid based antibiotic regime” by Gründer et al, candidate for publication in Antibiotics. Although the authors did an effort to address some of the points raised by this reviewer previously, my overall impression is that there are not much differences compared to the original manuscript. Actually, the content regarding results is more or less the same and my opinion is that the manuscript has not been further enriched. There are no comparisons with other treatment alternatives and no new data regarding linezolid treatment has been added. We do not know how patients failed; between first and two stage procedures? After the second stage? When did they fail during follow-up? 

Similarly, the structure of the results section has not been significantly modified; no presentation of the cohort in the first place, which can be found in Table 3 and 4, before failure data (not common this sort of presentation), for example. There are data in the results section that probably should be in the results section (Lines 262-264; Lines 269-271). Another example is that a mention to Kaplan Meier survival curves is still made in the Methods section, but there are no related figures in the manuscript, nor log-rank test analysis; has this really been done? Tables 1 and 2 perform a comparison between failed and non-failed patients but there are no p values for comparisons. Despite your statisticians’ recommendations, I still do not understand 95%CI when performing comparisons between failed and non-failed regarding covariates (not usually found in similar manuscripts).

In summary, my impression is that the authors can improve their manuscript, if major changes are made, but at present form, the manuscript is not suitable for publication. I encourage the authors to make significant changes in the manuscript.

Some sentences can be improved. 

Author Response

Dear reviewer,

thank you very much for taking time again to work on our manuscript. 

I fear that in some aspects, I did not get your point correctly and am trying to furhter clarify, explain and add to our findings.

Regarding the specific comments:

There are no comparisons with other treatment alternatives and no new data regarding linezolid treatment has been added

Thank you very much, I am not sure if we mean the same here. This is a retrospective cohort who were all treated with oral linezolid as part of a two-stage revision arthroplasty for PJI. All patients have the same diagnosis and underwent oral linezolid treatment as part of the systemic treatment as a mutual factor. There is no control group and therefore we can't provide data on treatment alternatives. To our knowledge, there are only studies on linezolid vs. other oral antibiotics in less resistant organisms where other antibiotics could be used as well.

I am unsure what data on linezolid treatment is missing in your opinion?

We do not know how patients failed; between first and two stage procedures? After the second stage? When did they fail during follow-up? 

We had added data on the definition of failure (Diaz-Ledema et al.) as well as data on microbiological failure during the revision. We have clarified that all failures occured after second-stage reimplantation in the results in line 76/77. Thank you for pointing that out, this aspect was missing. This line also gives the time (Median 7 months).

Similarly, the structure of the results section has not been significantly modified; no presentation of the cohort in the first place, which can be found in Table 3 and 4, before failure data (not common this sort of presentation), for example. There are data in the results section that probably should be in the results section (Lines 262-264; Lines 269-271).

We are happy to move these findings to the results section, however this is potentially against journal standards as we used the template that see the methods section as last section. I agree with you that it would be helpful to have the information from Tables 3 and 4 before reading the results. I have moved the tables up and added to the results

Another example is that a mention to Kaplan Meier survival curves is still made in the Methods section, but there are no related figures in the manuscript, nor log-rank test analysis; has this really been done?

Thank you very much, we have added figure 1 displaying the Kaplan-Meier survival with reinfection as the endpoint. The use of the log-rank test has been listed in the statistical methods section. The data on differences in survival in results section 2.5 are based on log-rank test calculations.

Tables 1 and 2 perform a comparison between failed and non-failed patients but there are no p values for comparisons.

Thank you , we have added the relevant factors with p-values in the results 2.5 section. We have also added them to tables 1 and 2.

Despite your statisticians’ recommendations, I still do not understand 95%CI when performing comparisons between failed and non-failed regarding covariates (not usually found in similar manuscripts).

Thank you very much, as i am not statistician myself i need to rely on their recommendations at times. These results were based on the log-rank test that compares Kaplan-Meier survival estimates. Some journals/reviewers/editor prefer to have isolated p-values while others only want the 95% CI in order to allow readers to the actual numbers themselves. As the CIs overlap and are quite wide in these comparisons, it shows that the findings are statistically fragile. If you wish, I am happy to only provide the p-value.

I hope that the changes made, qualify our paper for publication.

Kind regards and thanks again